# The Measure of Motion Similarity for Robotics Application [note 1]

**DOI:** 10.3390/s23031643

**Published:** 2023-02-02

**Authors:** Teresa Zielinska, Gabriel Coba

**Affiliations:** 1Faculty of Power And Aeronautical Engineering, Warsaw University of Technology, 00-665 Warsaw, Poland; 2Independent Researcher, Quito 17203, Ecuador

**Keywords:** motion synergy, human motion, trajectories comparison

## Abstract

A new measure of motion similarity has been proposed. The formulation of this measure is presented and its logical basis is described. Unlike in most of other methods, the measure enables easy determination of the instantaneous synergies of the motion of body parts. To demonstrate how to use the measure, the data describing human movement is used. The movement is recorded using a professional motion capture system. Two different cases of non-periodic movements are discussed: stepping forward and backward, and returning to a stable posture after an unexpected thrust to the side (hands free or tied). This choice enables the identification of synergies in slow dynamics (stepping) and in fast dynamics (push recovery). The trajectories of motion similarity measures are obtained for point masses of the human body. The interpretation of these trajectories in relation to motion events is discussed. In addition, ordinary motion trajectories and footprints are shown in order to better illustrate the specificity of the discussed examples. The article ends with a discussion and conclusions.

## 1. Introduction

The medical literature indicates that clinical assessment of gait and posture relies mainly on self-reports and simple observations for diagnostic purposes. Valuing such data is mostly subjective [1]. For medical research, however, deeper analyses are important, including the study of synergy in the human movement system (synergy: from the Greek word synergy, synergos—cooperation). Typical physiological movement synergies can be disturbed by physical injuries or diseases of the nervous system. Thus, pathologies may manifest themselves as additional synergies [2], or physiological synergies may be distorted [3,4]. In addition to diagnosing diseases and monitoring rehabilitation processes [5], synergy analysis provides knowledge about mechanisms of motor control [6]. Controlling smart prosthetics or exoskeletons [5,7] often uses electromyographic (EMG) data along with information about the synergy of muscle activity. Sometimes, EMG data are additionally combined with data from other sensors, which allows for more precise prediction of human movement [8].

Robotics scientists are studying animal movement synergies to translate them into robotic movement patterns. In our earlier research, we developed a motion pattern generator inspired by the biological central pattern generator (CPG). By selecting the appropriate parameters, the generator captured the synergy of movement in the human hip and knee joints [9]. The generator has been successfully implemented in the bipedal robot [10]. Knowledge of the coordinated movement of body parts also allows the development of simplified models of locomotion [11]. Inverted pendulum models with a rigid rod or double inverted pendulum models are used to describe the properties of slow gaits and to synthesize [12] controllers. The spring model of an inverted pendulum represents the fast gaits [13]. Simplified models help in designing the trajectory of the robot’s point mass and determining the steps, but they do not explain how to properly coordinate the movement of individual body parts. In a typical human gait, slight tilts of the trunk are coordinated with the movement of the legs and the swing of the upper limbs. This helps maintain dynamic postural stability. In the literature, kinematic synergies are defined as coordinated joint movements that in effect reduce the number of degrees of freedom whose movement must be defined [14]. The work [15] investigated the kinematic synergies of the leg joints. The article [16] explored velocity synergies in joints, named first-order synergies. In the work [17], the authors emphasized that an upright standing posture is maintained thanks to the coordinated interaction of the ankle and hip joints. This hypothesis was verified using a robot prototype.

The existence of synergies (or similarities) is mainly identified using distance measures. The most commonly used is the Euclidean distance between sets or data points (e.g., [18]). The work [19] used a PCA-based (Principal Components Analysis) approach to determine similarities in upper limb movements. PCA reduces the original multidimensional space of correlated sets of variables by projecting onto a smaller space of uncorrelated variables. Therefore, such projection indicates existing synergies. In addition, the correlation coefficients between dependent data [20,21] can be obtained. In [15], the PCA-based method was used to describe the kinematic synergy of human leg movement. Posture matrices containing movement data (joint angles) for each movement task were constructed, and synergies were identified as a result of the PCA projection. A similar approach was used in [21] to test hand movement. The state of the art indicates that synergy studies mainly concern periodic movements or repetitive activities. Moreover, such research is more popular in medicine than in robotics. The methods used rather do not allow for the detection of momentary synergies in non-periodic movements and their easy visualization in time. One of the frequently studied simple cases is inter-limb coordination in bi-manual tasks. Relative phase and relative time [22,23] are here used to describe motion similarities. The relative phase is calculated based on the phase difference of the limbs motion. Relative time is a measure of the temporal coordination of the limbs and is a measure of the difference in time of movement of these limbs. Other measures of inter-limb coordination are the coordination index, the coordination variability index, or other simple statistical measures such as variance or cross-correlation [24,25].

PCA methods are widely used for image processing (e.g., [26]). They work well to reduce the dimensionality of the data. However, the results given in the form of eigenvalues lose the intuitive aspect and do not allow for the analysis of temporary similarities, which is a disadvantage of PCA methods when studying movement similarities. Methods that determine relative time and relative phase are well suited to the study of periodic motions, but are not suitable for the investigation of momentary synergies in non-periodic motions. The coordination index and the correlation coefficient are only single measures. Therefore, based on the state of knowledge, it is concluded that there is no measure that would allow for easily comparing the movement of point masses of a human (or robot) in order to detect time synergies. To fill this gap, the measure described in this article was proposed.

## 2. Similarity Measures

In general, the similarity between curves or datasets is studied for many purposes and different measures of similarity are used. As already mentioned, the simplest approaches use the Euclidean distance metric. Modifications to the distance metric are also applied, such as normalized similarity (normS) [27], which, for two sets of samples *a* and *b*, is expressed as:(1)$$normS=\frac{1}{1+dist(a,b)}$$
where dist(a,b) in Formula (Equation 1) denotes the distance between *a* and *b*.

Another popular measure is distance correlation analogous to the Pearson correlation coefficient. The distance correlation (distCr) (Equation 2) is expressed as the distance covariance (${distCv}^{2}$) divided by the product of the distance standard deviations [20]:(2)$$distCr\left(a,b\right)=\frac{distCv^{2}\left(a,b\right)}{\sqrt{distVar^{2}\left(a\right)\;distVar^{2}\left(b\right)}},$$
where ${distVar}^{2}\left(\cdot \right)$ in (Equation 2) is the distance variance. Other correlation metrics, including kernel-based correlation metrics (such as the Hilbert–Schmidt Independence Criterion or HSIC), are also used to detect linear and nonlinear relations [28]. Another wide range of similarity analysis tools is offered by artificial neural networks, especially by clustering neural networks that group similar data using different distance metrics, e.g., [29].

## 3. Motivation and Objective

The study of synergies mainly focuses on comparing joint movement and analyzing EMG signals. Less attention is paid to the movements of point masses of the body. However, from the point of view of dynamic postural stabilization, the displacement of point masses [30] is of key importance. This fact is the motivation for the research described in this article. The aim was to develop a motion similarity measure that allows for an easy and graphical comparison of the displacements of point masses as a function of time. The overriding goal is to use this measure to synthesize the movement of humanoid robots, although the results may also be of interest to specialists in other fields (e.g., rehabilitation).

To illustrate the concept, the movement coordination of point masses of human body parts was examined. Two representative cases were studied: stepping forward and backward, and recovering a stable posture after an unexpected push from the side (arms free and fixed). This choice made it possible to compare slow dynamics (stepping) with fast dynamics (push recovery). The rest of the article is organized as follows—first, the measure of motion similarity is presented. The human movement data used to validate the measure are then described. Experiments are presented, and the obtained results are illustrated and commented on. Similarity measure trajectories are shown and discussed. The final part of the work contains a discussion and conclusions.

## 4. Formulation of the Motion Similarity Measure

Creating the similarity measure, the formula of Pearson’s [31] correlation coefficient was taken into account. The original formula is:(3)$$r\left(a,b\right)=\frac{\sum_{k=1}^{N}\left(a_{k}-\bar{a}\right)\left(b_{k}-\bar{b}\right)}{\sqrt{\sum_{k=1}^{N}{\left(a_{k}-\bar{a}\right)}^{2}\sum_{k=1}^{N}{\left(b_{k}-\bar{b}\right)}^{2}}}$$
where: r(a,b) is the Pearson’s correlation coefficient of the analysed variables, $a_{k}$ is the *k*-th sample from the first data set, $b_{k}$ is the *k*-th sample from the second data set, and $\bar{a}$, $\bar{b}$ are the mean values, respectively. The result r(a,b) belongs to the range 〈−1.00,+1.00〉, where:+1 means a perfect positive linear correlation between the variables;0 means that there is no linear correlation between the variables;−1 means a perfect negative linear correlation between the variables.

The correlation coefficient (Equation 3) is a measure of the linear relationship between two variables. This measure was used in our previous work [32]. Unfortunately, it produces a single score for the compared datasets. In order to determine the similarity measure for each moment of time, Formula (Equation 3) was modified, omitting the sum in the numerator and taking into account only the difference between the variable and its mean value:(4)$$r_{i}\left(a,b\right)=\frac{\left(a_{i}-\bar{a}\right)\left(b_{i}-\bar{b}\right)}{\sqrt{\sum_{k=1}^{N}{\left(a_{k}-\bar{a}\right)}^{2}\sum_{k=1}^{N}{\left(b_{k}-\bar{b}\right)}^{2}}}$$
where *i* denotes the *i*-th frame from the motion data record.

Equation (Equation 4) allows for analyzing similarities regardless of differences in amplitudes and shifts over a range of values (but not over a time). It can be applied in various combinations (depends on what is used as the reference) to study the motion of the human body point masses. In presented case studies, the ’base’ (reference) trajectory $r_{i}\left(b\left(CoM\right),b\left(CoM\right)\right)$ was formed considering displacements of point mass ($b_{i}\left(CoM\right)$) of the whole human body (*CoM*). It was compared with the similarity measure trajectories of the *CoM* and point masses of the body segments ($a_{i}$) that means $r_{i}\left(a_{i},b_{i}\left(CoM\right)\right)$. As $a_{i},\;b_{i}(CoM$, the x,y,z coordinates of masses were used accordingly.

The measure of similarity between the position of *CoM* ($b_{i}\left(CoM\right)$) and the position of some point mass of the body segment is expressed by:(5)$$r_{i}\left(a,b\left(CoM\right)\right)=\frac{\left(a_{i}-\bar{a}\right)\left(b_{i}\left(CoM\right)-\bar{b(CoM)}\right)}{\sqrt{\sum_{k=1}^{N}{\left(a_{k}-\bar{a}\right)}^{2}\sum_{k=1}^{N}{\left(b_{k}\left(CoM\right)-\bar{b(CoM)}\right)}^{2}}}$$

The similarity measure of *CoM* position related to itself:(6)$$r_{i}\left(b\left(CoM\right),b\left(CoM\right)\right)=\frac{\left(b_{i}\left(CoM\right)-\bar{b(CoM)}\right)\left(b_{i}\left(CoM\right)-\bar{b(CoM)}\right)}{\sqrt{\sum_{k=1}^{N}{\left(b_{k}\left(CoM\right)-\bar{b(CoM)}\right)}^{2}\sum_{k=1}^{N}{\left(b_{k}\left(CoM\right)-\bar{b(CoM)}\right)}^{2}}}$$

Let $r_{i}\left(a,b\left(CoM\right)\right)=r_{i}\left(b\left(CoM\right),b\left(CoM\right)\right)$, which, considering (Equation 5) and (Equation 6), is expressed by:(7)$$\begin{array}{l} \frac{\left(a_{i}-\bar{a}\right)\left(b_{i}\left(CoM\right)-\bar{b(CoM)}\right)}{\sqrt{\sum_{k=1}^{N}{\left(a_{k}-\bar{a}\right)}^{2}\sum_{k=1}^{N}{\left(b_{k}\left(CoM\right)-\bar{b(CoM)}\right)}^{2}}}=\\ \;\frac{\left(b_{i}\left(CoM\right)-\bar{b(CoM)}\right)\left(b_{i}\left(CoM\right)-\bar{b(CoM)}\right)}{\sqrt{\sum_{k=1}^{N}{\left(b_{k}\left(CoM\right)-\bar{b(CoM)}\right)}^{2}\sum_{k=1}^{N}{\left(b_{k}\left(CoM\right)-\bar{b(CoM)}\right)}^{2}}} \end{array}$$

Next, the difference is evaluated, which considering Equation (Equation 7) leads to:(8)$$r_{i}\left(a,b\left(CoM\right)\right)-r_{i}\left(b\left(CoM\right),b\left(CoM\right)\right)=\;\frac{\left(b_{i}\left(CoM\right)-\bar{b(CoM)}\right)}{\sqrt{\sum_{k=1}^{N}{\left(b_{k}\left(CoM\right)-\bar{b(CoM)}\right)}^{2}}}\left[\frac{\left(a_{i}-\bar{a}\right)}{\sqrt{\sum_{k=1}^{N}{\left(a_{k}-\bar{a}\right)}^{2}}}-\frac{\left(b_{i}\left(CoM\right)-\bar{b(CoM)}\right)}{\sqrt{\sum_{k=1}^{N}{\left(b_{k}\left(CoM\right)-\bar{b(CoM)}\right)}^{2}}}\right]=0$$

Assuming that $(b_{i}\left(CoM\right)-\bar{b(CoM)})\neq 0$, relation (Equation 8) is fulfilled when:(9)$$\frac{\left(a_{i}-\bar{a}\right)}{\sqrt{\sum_{k=1}^{N}{\left(a_{k}-\bar{a}\right)}^{2}}}=\frac{\left(b_{i}\left(CoM\right)-\bar{b(CoM)}\right)}{\sqrt{\sum_{k=1}^{N}{\left(b_{k}\left(CoM\right)-\bar{b(CoM)}\right)}^{2}}}$$

Based on (Equation 9), the similarity measure curves will be close to each other with a similar shape when the movement of the considered quantities will be similar towards their mean positions, divided by the summed squares of deviation.

## 5. Validation of the Measure: Materials and Methods

The measure of similarity has been validated using case studies. For this purpose, data recorded for different type of movement were used, namely for voluntary movement (stepping) and for movement forced by an unexpected event (recovery of postural balance after a push).

The subject of the study was a healthy young woman (Table 1). After obtaining her agreement, the movements were recorded using a professional motion capture system (VICON). The data concerned the positions of 30 markers attached to the body. Finally, the data from 22 markers were taken into account because it made a set sufficient to construct stick diagrams and for motion analysis. The publicly available 3D Motion Kinematic and Kinetic Analyzer (Mokka) software was used to visualize the position of the markers using the collected data.

The original data recorded by the VICON sensors were filtered, processed, and verified by system software (closed to the user). The system was located in the human movement diagnostics laboratory in the hospital; therefore, the stages of data registration and processing were subject to strict regulations regarding data quality and registration conditions. The MOKKA software associated with the system (see Figure 1) made it possible to visualize the markers and observe their trajectories. The processed data contained a lot of records in addition to the trajectories of the markers. Among other things, these were the angular trajectories of the body joints. Using anthropometric data, a three-dimensional stick diagram of the human body was elaborated in the MATLAB system. The stick diagram consisted of 11 segments:head and neck,left upper arm,right upper arm,left forearm,right forearm,trunk,pelvis,left thigh,right thigh,left shank and foot,right shank and foot.

Anthropometric data taken from [33,34] delivered the values and locations of a human body partial masses (see Table 2).

On the basis of the VICON data, an animation of human movement in MATLAB was obtained, and the motion trajectories of body point masses were calculated. The animation allowed the observation of movements at different speeds and by manipulating the viewing angle. The software also allowed for visualizing the footprints. We show them, in the next section, to better illustrate the relationship between motion events and human movement. Animation software, the history of footprints and human movement trajectories made it possible to verify the synergies indicated by the similarity measure.

In Figure 2, the segmented model of a human body with point masses indicated by the dots, and the reconstructed stick diagram are illustrated. The pelvis and spine position were reconstructed using the markers position that is indicated.

Referring to the three-dimensional illustrations presented in this article, it should be added that they are shown in perspective views, and the viewing angles are not always the same but have been chosen for each illustration separately to best illustrate the postures. Therefore, in different figures, a person can be perceived as a person of different height. All of these illustrations are for the same person with the data given in Table 1.

## 6. Case Studies

When conducting case studies, significant events were identified. Stick diagrams and footprints illustrating the analysed movements are used for illustration. Trajectories of similarity measures were determined. The results are demonstrated and commented on. In addition, for comparison, the motion trajectories are given.

### 6.1. Forward and Backward Stepping—Slow Dynamics

In the first case study, the data recorded for the person making a step forward and a step backward were used. The stepping began from normal standing posture when both legs are on the ground and next to each other.

In this stepping from a stand-still posture (double support), the following events were distinguished (Figure 3):A1 (0.80 s); the single support of the right leg begins, the left leg starts its transfer to the front;A2 (1.75 s); the single support of the left leg begins, the right leg starts its transfer to the front;A3 (2.70 s); the double support begins;B1 (6.75 s); the single support of the right leg begins when stepping back, the left leg starts its transfer to the back;B2 (7.90 s); the single support of the left leg begin, the right leg starts its transfer to the back;B3 (8.70 s); the double support begins.

For better illustration, Figure 4 shows footprints. The *CoM* trajectory presented here helps to understand the relationship between posture and motion events. The starting footprints are at the bottom of the figure. The footprints located in the upper part of the figure show the situation after taking a step forward, and the footprints at the bottom of the figure (slightly behind in relation to the initial position) demonstrate the situation after taking a step back.

Figure 5a shows the similarity measure obtained for motion along the X-axis. The black dashed line corresponds to the base curve created for the measure of auto-similarity (overall *CoM* with itself—rel. (Equation 6)), while the other curves calculated using (Equation 5) correspond to the other point masses.

Comparing Figure 6a containing the trajectories of displacements of point masses of the body with Figure 5a, one can see that, in the latter figure, the differences and similarities of displacements are much more visible.

The greatest similarity to the reference trajectory (obtained for *CoM*) is for the head, trunk, hips and arms. There is no similarity (i.e., no coordination with the reference trajectory for the right shank between events A1 and A2, and for the left shank between events B2 and B3. In the first case, it is the time interval when the left leg moves forward, and in the second case, when the right leg moves backward. Bearing in mind that in both these cases either the right leg or the left leg is supporting the body and the foot is stationary while the rest of the body is moving (at the slightest displacement of the shank of the supporting leg), the diagrams reflect the real situation well. A closer look at the graphs, and in particular at the trajectories obtained for the left and right shank and right arm, reveals lateral tilts. In the case of transferring the right leg forward (A2–A3) and the left leg backward (B1–B2), the trajectories of the point masses of the thigh and shank of the left and right leg, respectively, show mutual coordination, but they differ from the reference trajectory.

When analyzing the similarity measures obtained for the displacements of point masses along the Y axis (Figure 5b), it can be concluded that the variability of all trajectories is similar, which means that all displacements are coordinated along the motion direction. A slight shift and slightly smaller values towards the reference trajectory $r_{i}\left(y\left(COM\right),y\left(CoM\right)\right)$ for the left shank in the interval A1–A2 result from the fact that the left leg (leg just starting the forward transfer) and especially its shank are behind the trunk. A similar situation applies for the shank of the right leg just before starting its back transfer in interval B2–B3. In the A2–A3 interval, the transfer of the right leg is not so visible. The smaller values of the shank trajectory of the transferred leg towards the reference trajectory are observed in the interval B1–B2.

The greatest differences in similarity measures for displacements along the Z axis (Figure 5c) concern both arms and legs, but in the case of upper limbs (arms and forearms), the similarity to the reference trajectory is observed, which proves the coordination of displacements along the vertical axis. The differences are seen for the thigh and shank of the left leg in interval A1–A2, which is caused by the fact that this leg is transferred, i.e., its movement is different from the movement of the rest of the body—this is especially evident for the shank. The lack of coordination with the reference trajectory for the left shank is moreover visible at the beginning of the A3 interval. This is intuitively unexpected because, when the right leg starts the transfer, the left leg is one ground. It results from a slight bending of the supporting leg (left leg). The trajectory of the similarity measure for the right shank has in this time interval clearly bigger values than the reference trajectory.

The presented results show that, unlike the usual trajectories (Figure 6), the introduced measure facilitates the visual identification of temporary synergies in the movement of point masses. The analysis of ordinary trajectories is more troublesome to find similarity/dissimilarity in motion. However, trying to analyse these trajectories, it can be concluded that the greatest differences in relation to the *CoM* trajectory can be observed for the X displacement of the left and right shank in the intervals A1–32 and B1–B3. For movement along the Y direction, the differences are not so obvious, and for the Z direction, again, the differences are most noticeable for the left and right shank.

### 6.2. Recovery after Side Push (Free Arms)—Fast Dynamics

The person received an unexpected but light push by the force applied to the right shoulder (Figure 7).

After the push, the person ’rejects’ the disturbance caused by the push. The body parts are involved to regain the postural stability without any delay. First, the reactive motion is performed by the left leg to avoid falling and next the person resumes the posture by moving the right leg to the original position. In this case, the most relevant but short fragment of action, which is regaining the postural stability, was analysed. The following events were distinguished:B1 (6.25 s); double support, the person is pushed to the left from the right side;B1–B2 (6.25–6.7 s); single support by right leg, the person reacts to the push by moving slightly the left leg up and transferring it to the direction of the push in order to gain equilibrium, by the end of this interval, the left leg moves down;B2 (6.7 s); the left-leg touches the ground;B2–C1 (6.7–7.48 s); single support by the left leg, the right leg is raised up and moves to the left;C1 (7.48 s); the body is in a maximum side sway to the left, the posture recovery begins, and the left leg moves to the right side.
In the moment 8.10 s, a lateral right-leg single support begins. and the left leg moves towards the right leg. The final stage is double support phase.

Good coordination of the displacements of point masses along the X axis (push direction) can be noticed (Figure 8a). Only after the push do the right forearms show less coordination of movement in relation to the reference trajectory $r_{i}\left(x\left(COM\right),x\left(CoM\right)\right)$. Small differences in the behavior of the forearms and shanks are also visible closer to the moment C1, when the right leg is shifted towards the left leg to regain the postural equilibrium.

The measures of similarity for the displacements along the Y axis (Figure 8b) reflects the process of equilibrium recovery well. The trajectories of similarity measures for the upper body and pelvis are close to the reference trajectory $r_{i}\left(y\left(COM\right),y\left(CoM\right)\right)$, although here, too, some differences can be noticed for the trajectory of the pelvis just after the push. Throughout the entire time interval, the similarity measure trajectories are mutually similar for the arm and forearm of the left limb and the arm and forearm of the right limb, respectively, but they are different for each limb and differ from the reference trajectory. Thus, it indicates that each upper limb independently supports the process of regaining balance without changing its configuration in the XY plane. It can be also noticed that, in the first phase of the movement, the left upper limb makes more voluminous movements (there are greater differences comparing to the reference trajectory), and in the second phase—it is a case for the right upper limb. The side shift of the left leg in the B1–B2 time interval is coordinated with the *CoM* movement (the trajectories of the thigh and shank of the left leg are close to the reference trajectory) and placing the right leg next to the left restores the coordination with the reference trajectory (in the vicinity of the C1 point).

The similarity measures for the motion along the Z direction (Figure 8c) shows that the up and down independent movements of the body parts are relevant for recovery of postural stability. The similarity to $r_{i}\left(z\left(COM\right),z\left(CoM\right)\right)$ is only in some short intervals and for some point masses. A greater mutual similarity (then the similarity to the reference trajectory) can be noticed for the left forearm and arm, and for right arm and forearm, which means (together with closer look to *x* and *y* results) that the hands rather do not bend at the elbows but by swinging help to recover the postural stability.

For the comparison, the displacement trajectories are given in Figure 9. The conclusion about similarity in the motion trends is here even more difficult than in the case of stepping. The biggest differences comparing to the *CoM* motion characteristics are within the B1–B2 period up to C1 for left and right shanks and left and right forearms; however, it is not so easy to deliver more precise conclusions.

### 6.3. Recovery after Side Push (Tied Arms)—Obstructed Fast Dynamics

The person received an unexpected light push such as in the previous case, but now the arms were tried to the trunk (Figure 10).

The lack of arms movement is ’compensated’ by taking a greater step to the side to regain postural stability. As before, the reactive movement involves moving the left leg to the left to avoid falling. Then, the person returns to the position by moving the legs back to their original position. As before, only a short fragment of regaining postural stability was studied with the following events:B1 (4.6 s); double support—the person is pushed to the left from the right side,B1–B2 (4.6–5.15 s); single support by right leg—the person reacts to the push by moving slightly the left leg up and transferring it to the direction of the push in order to gain equilibrium; by the end of this interval, the left leg moves down;B2 (5.15 s); the left-leg touches the ground;B2–C1 (5.15–5.5 s); single support by the left leg—the right leg is raised up and moves to the left;C1 (5.5 s); the body is in a maximum side sway to the left, the posture recovery begins, and the left leg starts moving back to the right side.
In the time 6.10 s, a lateral right-leg single support begins, and the left leg moves towards the right leg. The final stage is double support. Figure 11 is illustrating similarity measure trajectories. In general, all similarity trajectories for this case show the same trends as for the case of pushing the person with free arms.

The motions of point masses along the X axis (along push direction), Figure 11a is strongly coordinated, and all similarity trajectories are quite close. Since the arms are tied to the trunk, there is coordination of the forearms, unlike in the previous case.

The measures of similarity of displacements along the Y axis (Figure 11b) clearly indicate that the arms and forearms are not free and move in the same way as *CoM*. Small differences from a reference trajectory at the beginning of the examined period may result from the slight relative displacement of arms towards the trunk, since tying the hands was not absolutely rigid. The trajectories of the thigh and shank illustrate the process of a side stepping well (as in the previous case). The lateral shift of the left leg in the time interval B1–B2 is coordinated with the movement of the *CoM*, which means that the similarity trajectories of the left leg thigh and shank are close to the reference trajectory, but not as close as for the free-handed case. It is worth noting that the head trajectory is slightly different compared to the previous case. The movement of the head along the direction of the push is less coordinated with the *CoM* and the trunk position as well. It is to be expected that this is due to the difficulties in regaining postural equilibrium.

Similarity measures along the Z direction (vertical) (Figure 11c) confirm immobilization of the hands relative to the trunk, and their trajectories are close to the reference trajectory. For the right thigh and shank, greater values can be observed than in the previous case, which, together with the results obtained for the movement along the X and Y axes, indicate that the right leg contributed to a greater and independent extent to regaining stability. In the case of the head, greater coordination with the reference trajectory is observed than in the previous case.

The displacement trajectories for this case are shown in Figure 12. Looking at the displacements along the X axis, it is difficult to find differences in the motion trends. For the movements along the Y axis, the displacement of the left thigh is mostly different from the movement of *CoM* and the other masses, and for displacements along the Z axis, this applies to thigh and shank.

## 7. Discussion

It is obvious that other similarity measure propositions could be developed. As already mentioned, the simplest measure is distance between the points of the compared trajectories. Such a measure, however, ignores the overall variability and the mean value, which are specific to each movement. It means that distance differences are sensitive to ranges of motion. For example, completely different distances between the position of the hand masses and the *CoM* will be obtained when walking with the arms above the head and walking with the arms down.

The proposed formula was created with reference to Pearson’s correlation coefficient [31], but it should not be interpreted as a statistical measure. Pearson’s coefficient indicated important components, namely the deviation from the mean value and the sum of squared deviations from the mean values. Inclusion of these terms to the measure allows for considering the character of movements. For example, if the sum of deviations obtained for the considered point mass is relatively large, compared to the analogous term obtained for *CoM*, then, according to (Equation 9), the comparison will indicate differences even if the deviations will be locally equal. This means that for highly variable displacements of one quantity and small—for the other, the similarity trajectories will be shifted in value. However with similar local deviations, they will have a similar shape. Including the sums of deviations as a normalizing factor allows for finding similar trends in highly variable data. The proposed concept was considered simple and sufficient enough.

The general formula for the measure of similarity is given by (Equation 4). Its use is illustrated implementing the formulas (Equation 5) and (Equation 6) where *CoM* was chosen as the ’reference’ component. This choice resulted from several reasons. First, *CoM* is the main element in simplified dynamics analyses. Mass times acceleration provides force, and force multiplied by the acting arm is torque, which is a fundamental parameter in dynamic stability studies [30]. Secondly, the dynamic displacements of partial point masses contribute to the dynamics of the entire *CoM*, and such contribution may or may not be synergistic. Examples of synergy include arm swings, leg movements and *CoM* trajectory during normal walking (e.g., [35]). *CoM* was used as a normalizing factor in the sense presented in the section **Formulation of the motion similarity measure**. The description of the results of the case studies demonstrated that similarity measures obtained for different point masses can be compared with each other. The user can, of course, choose not *CoM* but another mass as the basis for the calculation—it depends on the research goal. The case studies are for one person only and are only illustrative of how the synergy measure can be used. It cannot be assumed that the results obtained define standards. It is obvious that people of different physiques and ages, suffering from different diseases and injuries, will have different body dynamics. Therefore, the presented cases cannot be generalized to the entire population. However the indicated synergies are logic taking into account the character of investigated movements and the postural stabilization. When examining synergies, we are mainly interested in determining which parts of the body move in a similar way. Accurate comparisons are not expected. This point of view is useful for humanoid motion design and for certain diagnostic tests of the patients. Comparison of old data with new data may indicate the disappearance of synergies or the emergence of new ones, which, as already mentioned, supports the diagnosis of diseases.

It is worth adding that we use the synergy study results in our research on the application of simplified body models to design the motion of humanoid robots. These results help to design the limbs displacement when the model of motion is known only in the form of reconfiguration of a double inverted pendulum [36].

Compared to the results obtained using other approaches, it should be emphasized that the developed method of similarity analysis gives the result for each instant of time, and allows for easily identifying similarities in the movement of point masses of the body, which is not a common feature of other methods described in the literature. In our other works [32], we analysed the trajectories of point masses of a body using a correlation coefficient. The goal was to replicate the movement of the human body using a double inverted pendulum. The body was thus divided only into upper and lower parts. A correlation coefficient (Equation 3) was determined between the displacements of the point masses of these parts of the body. The results showed a high correlation for walking and running with free and tied hands. This work was needed for a specific purpose, but also confirmed that evaluation of synergies using a single value is insufficient. It should be added that works on synergy include the classification of human posture, i.e., recognizing what posture a person assumes in a given period of time. Our team is also investigating this issue [37]. In this case, the goal is different from the goal of the measure introduced in this article. The results are given as semantic posture labels (e.g., standing posture, walking posture, crouching posture) and enable the recognition of human actions as posture sequences. Classifying neural networks is a typical tool used to recognize postures. Referring to the approaches most similar to the presented research, i.e., approaches focusing on comparison of the trajectories of movement, Dynamic Path Warping (DPW) should be mentioned [38,39]. In [38], human foot motion trajectories were compared with each other. DPW enables the determination of the similarity between two data sequences that differ in the rate of changes. DWP processes (warps) the compared signals (data sequences), making them independent of time and insensitive to acceleration and deceleration. Distance matrices and the so-called warping paths are then created to illustrate the similarity of the compared data. In [39], DPW was used to compare postures during rehabilitation exercises.

## 8. Conclusions

The proposed measure of motion synergy allows for:Visualization of relations between displacements of masses of body parts;Making it easier to identify movement synergies;Unlike Pearson’s coefficient, giving a result for every moment of time, i.e., it illustrates the similarities/differences of motion in the time domain.

The main contribution of the work is as follows:Introducing the formula for the similarity measure;Explanation of its meaning based on simple mathematical considerations;Demonstrating its use (interpretation) using various examples of slow and fast dynamics of non-periodic motions.

To justify the usefulness of the introduced measure, the case studies (stepping and recovery after a side push with arms free and tied) were presented. The trajectory of the similarity measure of the resultant body mass point (CoM) was used as the reference trajectory. The similarity measure indicated differences between the coordination of self-controlled movements (stepping example) and not fully controlled movements (push reactions).

The next stage of research on the similarity of human body movements should be to extend the analysed examples and examine the repeatability of the results. Bearing in mind the latter issue, it should be noted that human movement was recorded in a hospital diagnostic laboratory. In addition to the VICON system, which is used to diagnose patients, and thus must guarantee high accuracy, professional software was used to eliminate reading errors. The data can therefore be considered sufficiently accurate. Nevertheless, the study used the data of one person. Therefore, this article does not pretend to formulate general conclusions regarding the coordination of body movements, but rather demonstrates the information that the introduced measure of similarity may potentially carry. The presented case studies justified that the proposed measure facilitates easy identification of temporary synergies of displacements of individual body parts.

## Figures and Tables

**Figure 1 sensors-23-01643-f001:**
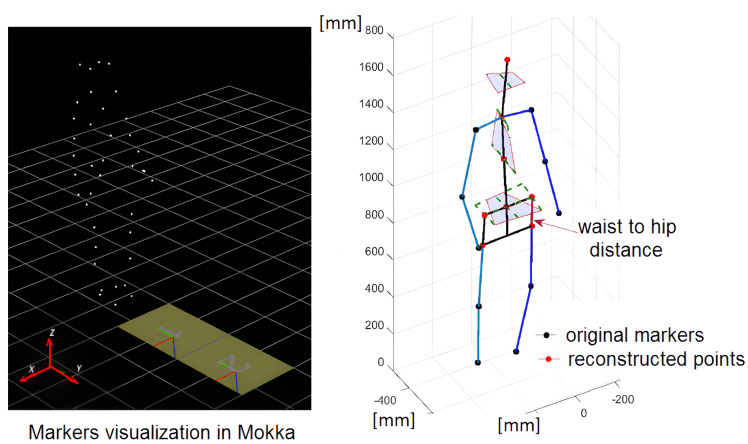
Visualization of markers in Mokka software (**left**) and reconstructed stick diagram of a human body (**right**).

**Figure 2 sensors-23-01643-f002:**
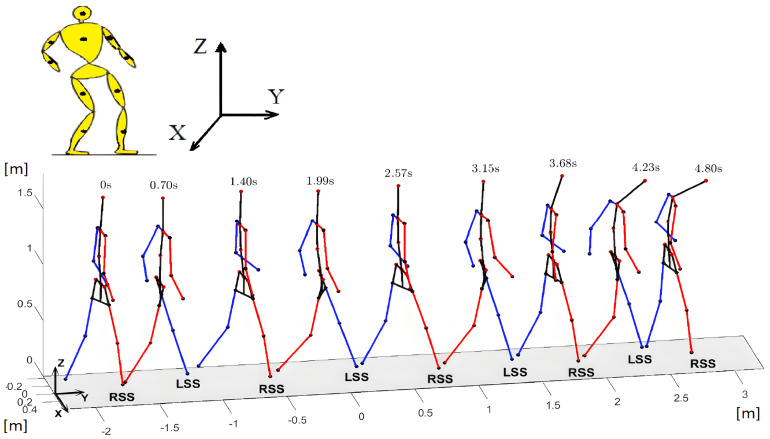
Illustration of considered body segments and example illustration of the sequence of stick diagrams during the walk. RSS denotes a single support phase of the right leg and LSS—single support phase of the left leg.

**Figure 3 sensors-23-01643-f003:**
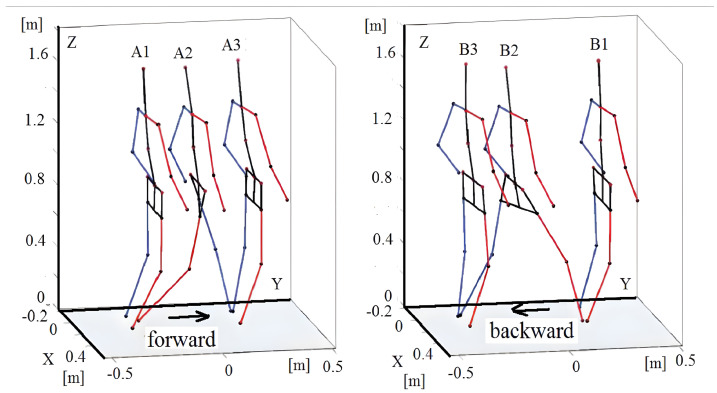
Visualization of events during forward and backward stepping.

**Figure 4 sensors-23-01643-f004:**
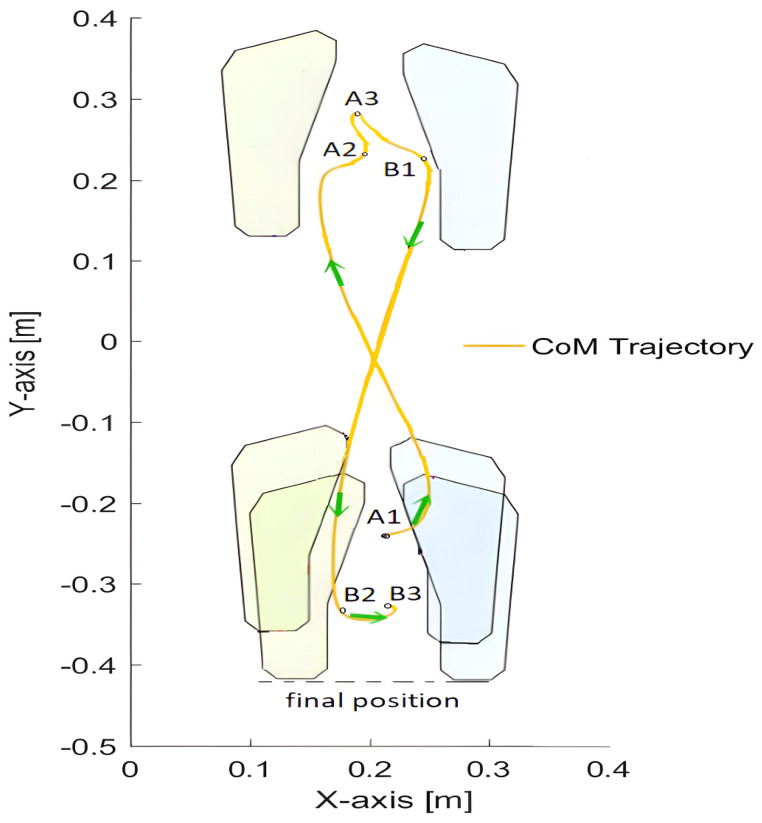
Footprints illustrating the stepping. The yellow curve denotes the *CoM* trajectory, and motion events are indicated.

**Figure 5 sensors-23-01643-f005:**
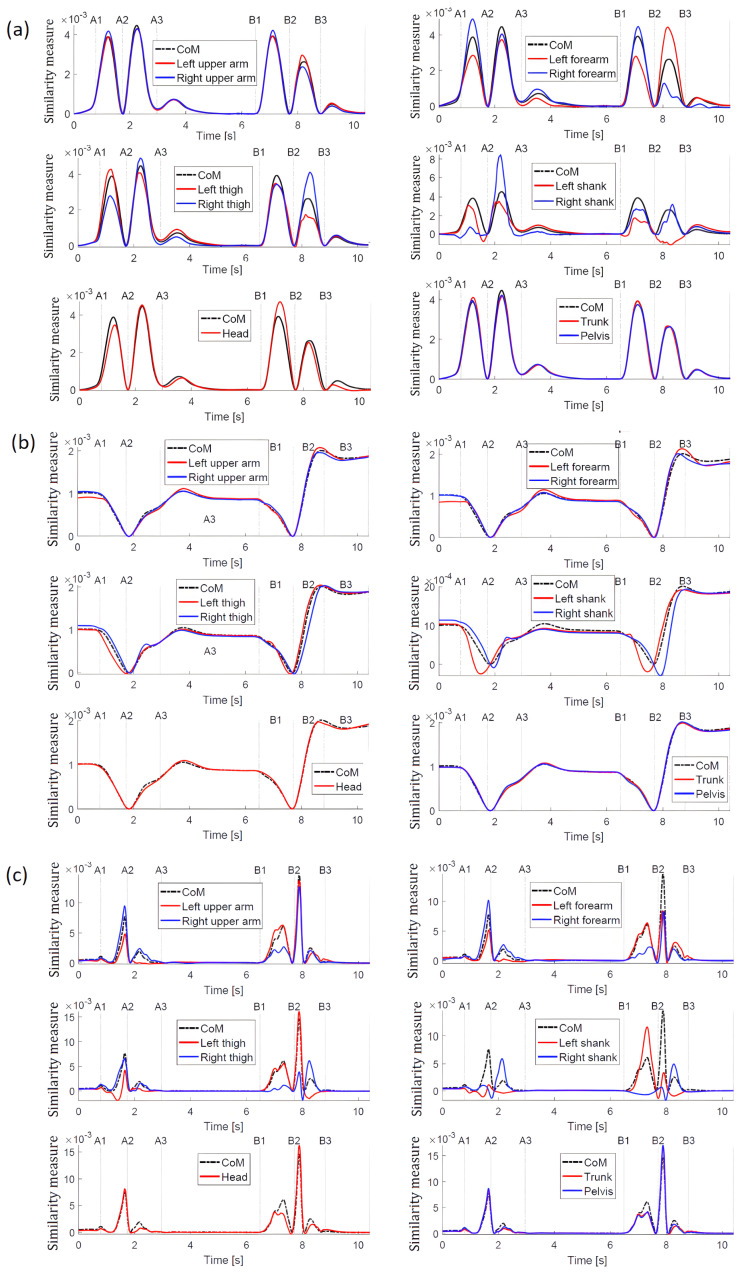
Trajectories of the similarity measure during the stepping. (**a**) trajectories along the X-axis; (**b**) trajectories along the Y-axis; (**c**) trajectories along the Z-axis.

**Figure 6 sensors-23-01643-f006:**
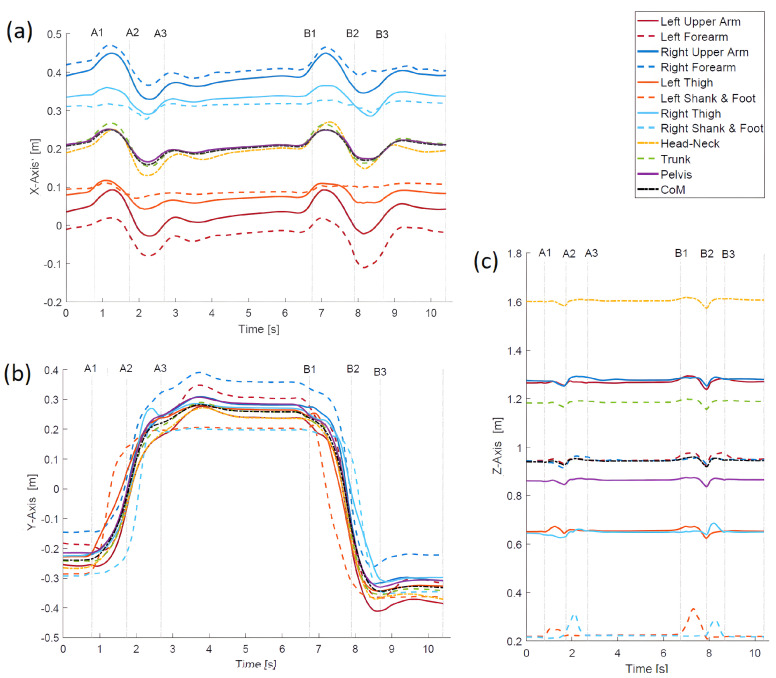
Displacement trajectories during stepping. (**a**) trajectories along the X-axis; (**b**) trajectories along the Y-axis; (**c**) trajectories along the Z-axis.

**Figure 7 sensors-23-01643-f007:**
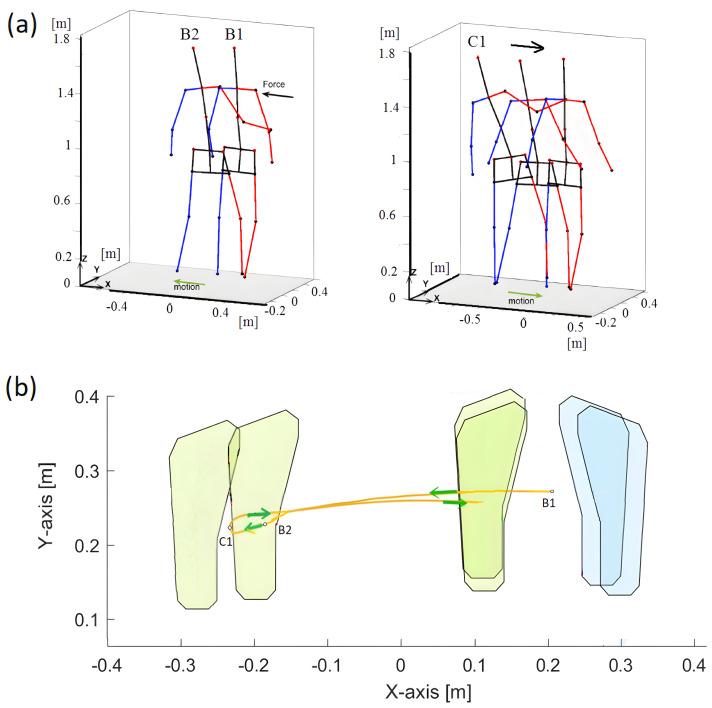
Push recovery with free arms. (**a**) visualization of events; (**b**) footprints illustrating the push recovery. The yellow curve denotes the *CoM* trajectory; motion events are indicated.

**Figure 8 sensors-23-01643-f008:**
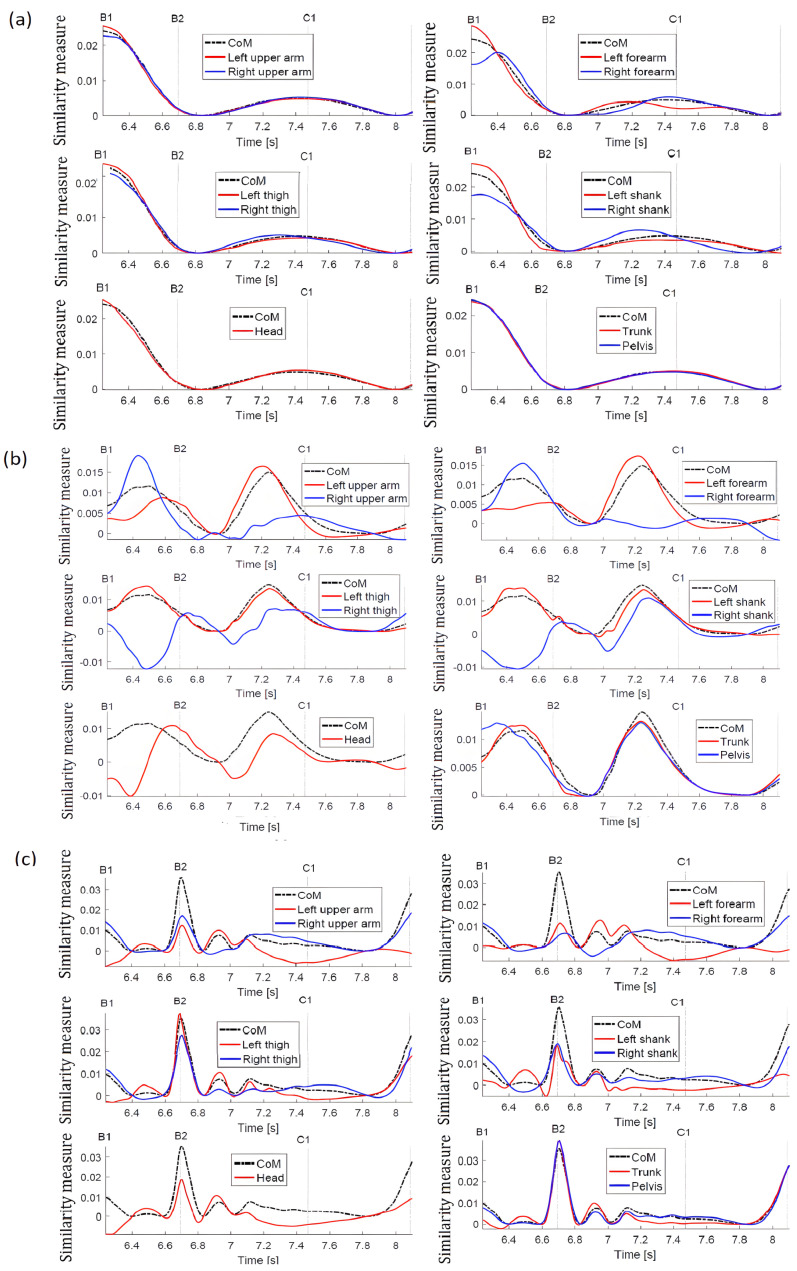
Trajectories of the similarity measure during the push recovery (free arms). (**a**) trajectories along the X-axis; (**b**) trajectories along the Y-axis; (**c**) trajectories along the Z-axis.

**Figure 9 sensors-23-01643-f009:**
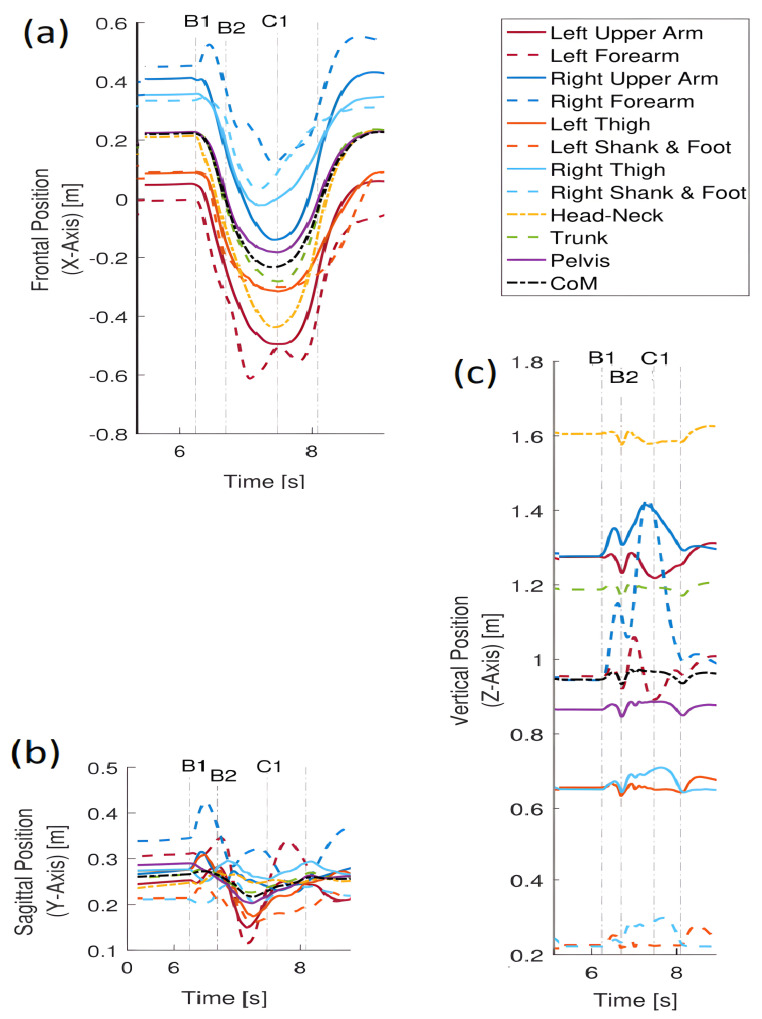
Displacement trajectories during push recovery with free arms. (**a**) trajectories along the X-axis; (**b**) trajectories along the Y-axis; (**c**) trajectories along the Z-axis.

**Figure 10 sensors-23-01643-f010:**
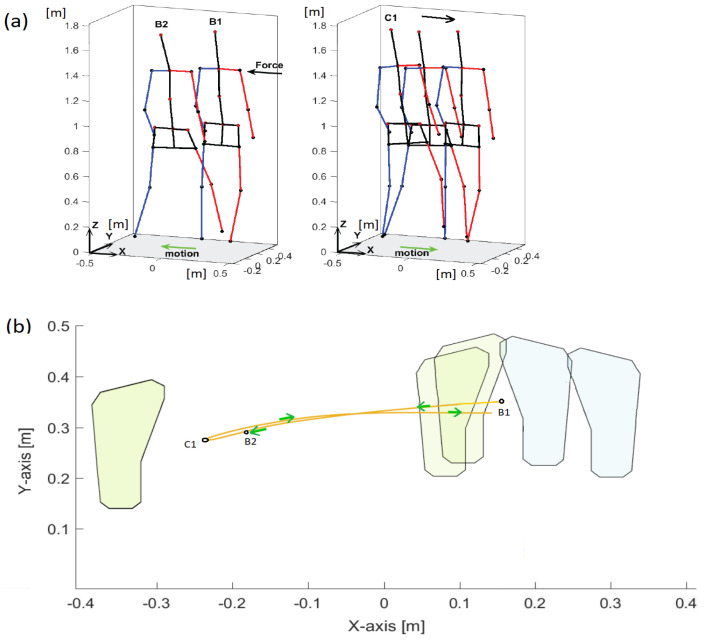
Push recovery with tied arms. (**a**) visualization of events; (**b**) footprints illustrating the push recovery. The yellow curve denotes the *CoM* trajectory, and motion events are indicated.

**Figure 11 sensors-23-01643-f011:**
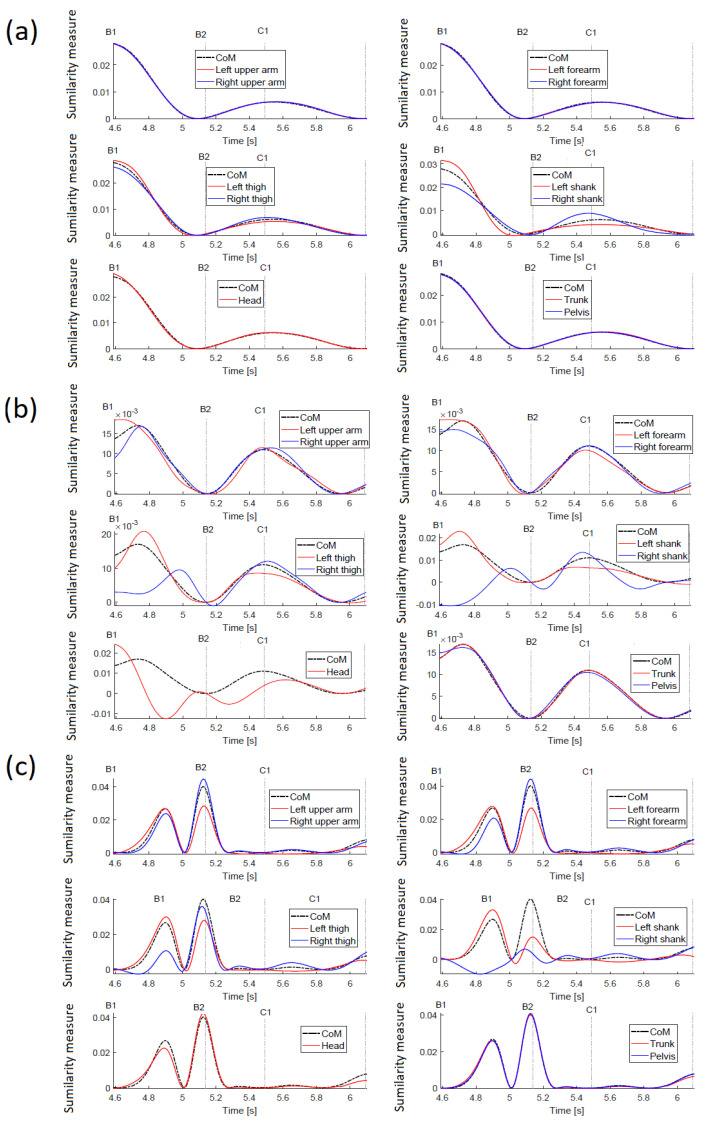
Trajectories of the similarity measure during the push recovery (tied arms). (**a**) trajectories along the X-axis; (**b**) trajectories along the Y-axis; (**c**) trajectories along the Z-axis.

**Figure 12 sensors-23-01643-f012:**
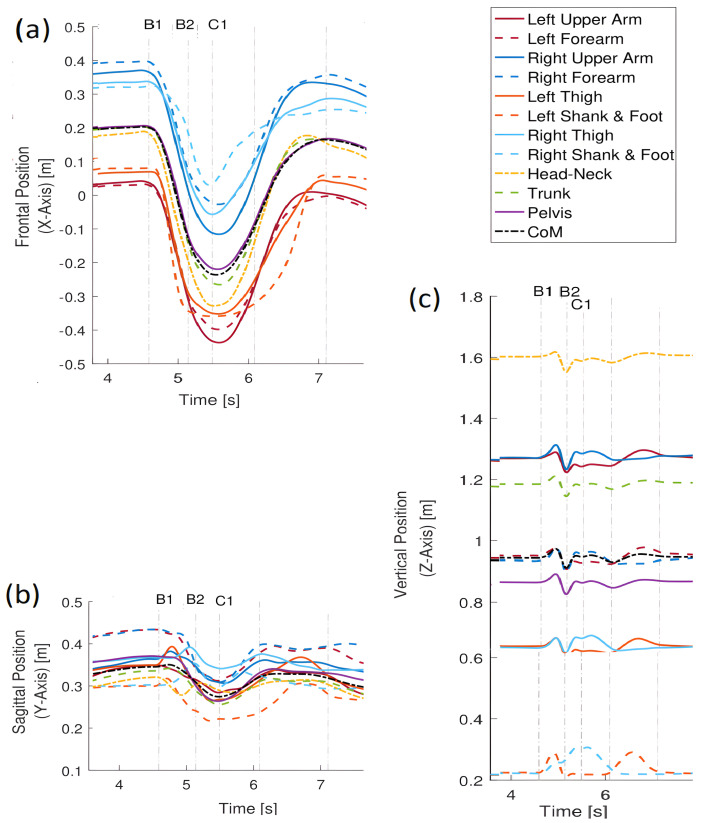
Displacement trajectories during push recovery with tied arms. (**a**) trajectories along the X-axis; (**b**) trajectories along the Y-axis; (**c**) trajectories along the Z-axis.

**Table 1 sensors-23-01643-t001:** The subject data.

Parameter	Value
Gender	Female
Mass [kg]	56
Height [cm]	168
Percentile	80-th
Waist to hip distance [cm]	16.63

**Table 2 sensors-23-01643-t002:** The anthropometric data.

Segment	Mass [kg]	*CoM* Location from Distal Segment
Upper arm	1.56	0.436
Forearm and hand	2.232	0.682
Thigh	5.6	0.432
Head and neck	4.536	0.646
Shank and foot	3.416	0.606
Trunk	19.88	0.630
Pelvis	7.952	0.105

## Data Availability

Source data are available upon request and subject to approval by the Kinesiology Laboratory.

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
