# Peer review of "The Measure of Motion Similarity for Robotics Applicationâ€"

_sensors, 2023, doi:10.3390/s23031643_

Round 1
Reviewer 1 Report
1. problem statement is not clear. Please provide a clear gap in the introduction section.
2. Almost all the equations used are not cited in the text.
3. Most of the equations used are already presented by the researchers. I think the authors should properly reference them.
4. More discussion is needed on the results presented. The authors must compare the results with the previously presented research.
5. The conclusion must be given point-wise highlighting the main contributions of the work.
Reviewer 2 Report
In this paper, the authors proposed a measurement of motion similarity with visual display by comparison the movement of point masses of the body. Two cases, stable walking and push recovery were studied to illustrated the prosposed measurement. Tests were carried on to domenstrate the process and results of similarity measurement. Below are some suggestions and comments.
1. Is there only one subject? The size of the subject shown among the figures are obviously different.
2. The center of gravity is calculated by all the parts of the body. It is obvious that each part will not be consistent with the center of gravity during movement, so what is the point of using the center of gravity as a reference point?
3. Please check the format and typing carefully. The font color of the title of the Figure 4 and 6 should be black. And Principsl should be Principal
Reviewer 3 Report
The study of the synergy of human movements is needed in various research areas. Unusual synergies may signal neurological disorders or vice versa. The synergies are expressed in various ways, but there are no indicators that could easily show temporary synergies in the movement of body parts. Hence, this manuscript proposes a new measure of motion similarity. On the whole, the research content of this manuscript is relatively meaningful, but there are also some problems that need to be modified.
1. The abstract is an important part of an article. The abstract part of manuscript does not elaborate the main research contents, methods, technologies or means of the manuscript. the experimental results are not given, which will confuse the readers. It is suggested to reorganize the contents of abstract.
2. References cited in the text of the manuscript were not arranged in accordance with the requirements of academic papers. In academic papers, the order in which references are cited in the text of the manuscript starts from 1 and increases progressively.
3. In the manuscript, "as shown in Figure/Table XX" should be given first before the figure or table to be displayed can be given, otherwise, it is not allowed.
4. The horizontal and vertical units in some figures are not given, such as Figures 1, 2, and so on.
5. The writing format of the picture’s illustration in the manuscript does not meet the requirements of the journal format. For example, “Figure 4. Stepping. foot prints”.
6. In Section 5, the validation of the measure, materials and methods are respectively given. However, the verification environment is relatively general, which makes it difficult for readers to understand whether the verification is carried out by means of simulation or real experiment. If it is a real human walking experiment, can you provide relevant information such as the human body picture to increase the persuasiveness of the article.
7. The human body is quite different, and people of different ages, genders, etc. have different differences. This manuscript gives few verification cases. Whether it can truly verify the content of this manuscript remains to be considered. It is suggested that batch classification is to add sample cases to prove the method proposed in this manuscript.
Round 2
Reviewer 1 Report
Improvments have been made.
Reviewer 3 Report
The authors of this manuscript have revised the manuscript in detail according to the comments of the reviewer to make it more perfect. The reviewer believes that the manuscript has reached the preliminary acceptance standard. It is important to note that the reviewer found that in the current manuscript, the reference of formulas in the text is not standard and uniform. In general, the formula reference in the text should be Equation (X) or Eq. (X), for example Equation (2) or Eq. (2).